# RXRs control serous macrophage neonatal expansion and identity and contribute to ovarian cancer progression

María Casanova-Acebes [1,2,3,7], María Piedad Menéndez-Gutiérrez [4,7], Jesús Porcuna [4], Damiana Álvarez-Errico[4], Yonit Lavin[1,2,3], Ana García[4], Soma Kobayashi[1,2,3], Jessica Le Berichel[1,2,3], Vanessa Núñez[4], Felipe Were[5], Daniel Jiménez-Carretero[6], Fátima Sánchez-Cabo[5], Miriam Merad[1,2,3 ✉] & Mercedes Ricote [4 ✉]

Tissue-resident macrophages (TRMs) populate all tissues and play key roles in homeostasis, immunity and repair. TRMs express a molecular program that is mostly shaped by tissue cues. However, TRM identity and the mechanisms that maintain TRMs in tissues remain poorly understood. We recently found that serous-cavity TRMs (LPMs) are highly enriched in RXR transcripts and RXR-response elements. Here, we show that RXRs control mouse serous-macrophage identity by regulating chromatin accessibility and the transcriptional regulation of canonical macrophage genes. RXR deficiency impairs neonatal expansion of the LPM pool and reduces the survival of adult LPMs through excess lipid accumulation. We also find that peritoneal LPMs infiltrate early ovarian tumours and that RXR deletion diminishes LPM accumulation in tumours and strongly reduces ovarian tumour progression in mice. Our study reveals that RXR signalling controls the maintenance of the serous macrophage pool and that targeting peritoneal LPMs may improve ovarian cancer outcomes.

[1] Department of Oncological Sciences, Icahn School of Medicine at Mount Sinai, New York, NY, USA. [2] Precision Immunology Institute, Icahn School of Medicine at Mount Sinai, New York, NY, USA. [3] Tisch Cancer Institute, Icahn School of Medicine at Mount Sinai, New York, NY, USA. [4] Area of Myocardial Pathophysiology, Centro Nacional de Investigaciones Cardiovasculares (CNIC), Madrid, Spain. [5] Bioinformatics Unit, Centro Nacional de Investigaciones Cardiovasculares (CNIC), Madrid, Spain. [6] Cellomics Unit, Centro Nacional de Investigaciones Cardiovasculares (CNIC), Madrid, Spain. [7]These authors contributed equally: María Casanova-Acebes, María Piedad Menéndez-Gutiérrez. ✉email: miriam.merad@mssm.edu; mricote@cnic.es

Macrophages are myeloid-derived cells that populate all tissues, where they contribute to tissue remodelling and protection against pathogens and injury[1]. Macrophages are heterogeneous and derive from two main lineages. Tissue-resident macrophages (TRMs) arise mainly from embryonic precursors[2] and reside in tissues for prolonged periods, whereas blood-derived macrophages are found mostly in injured tissues[3]. In a previous analysis by RNA-seq, ChIP-seq and assay for transposase-accessible chromatin with sequencing (ATAC-seq) of purified TRMs from six organs, we showed that the TRM epigenetic and transcriptional programme is unique to each tissue and is shaped by the tissue microenvironment in which TRMs reside[2,4]. For example, the ability of splenic marginal zone macrophages to trap circulating particulates and engulf marginal zone B cells is controlled by liver X receptor alpha (LXRα)[5], surfactant clearance by alveolar macrophages is facilitated by expression of peroxisome proliferator-activated receptor gamma (PPARγ)[6] and iron recycling and erythrocyte phagocytosis by splenic red pulp macrophages depends on Spi-C expression[7].

The peritoneal and pleural cavities are small fluid-filled spaces that contain a large population of immune cells, including T and B cells, mastocytes, dendritic cells, monocytes and macrophages[8]. In mice, the peritoneal and pleural cavities contain two macrophage subsets distinguished phenotypically by their size and differential expression of F4/80 and MHC class II (MHCII)[9]. F4/80^HI MHCII^LO large peritoneal macrophages (LPMs) are the most abundant population in the steady-state peritoneal space. LPMs have a typical macrophage morphology, including abundant cytoplasmic vacuoles and a high capacity to phagocytose apoptotic cells[9,10], and contribute to the maintenance of intestinal microbial homeostasis by promoting the production of IgA by gut B1 cells[11]. The small peritoneal macrophage (SPM) subset expresses lower F4/80 and higher MHCII levels and predominates after injury associated with infection or inflammation[9], playing important roles in bacterial removal and antigen presentation[8,9,12]. LPMs arise mainly from embryonic precursors that are recruited to the peritoneum prior to birth, and are essentially maintained locally through self-renewal[13,14], although monocytes slowly and continuously contribute to the LPM pool[15–17]. LPM maintenance is crucially dependent on the retinoic acid-dependent transcription factor GATA-6[11,18,19]. In contrast, SPMs differentiate from circulating monocytes, in a process reliant on IRF4[20].

Retinoid X receptors (RXRs) are members of the nuclear receptor superfamily of ligand-dependent transcription factors. They act as regulators of gene expression, exerting pleiotropic transcriptional control over a wide range of genetic programmes, including cell differentiation and survival, immune response, and lipid and glucose metabolism[21,22]. There are three RXR isotypes, RXRα (NR2B1), RXRβ (NR2B2) and RXRγ (NR2B3), which all show tissue-specific expression. These transcription factors form heterodimers with many other members of the nuclear receptor superfamily and also function as transcriptionally active homo-dimers[23]. RXR ligands include the vitamin-A derivative 9-cis-retinoic acid (9-cis-RA) and several endogenous fatty acids[21]. RXRs are important regulators of macrophage biology, playing key roles in inflammatory and autoimmune disorders and in bone homeostasis[22,24–26].

We recently identified RXRα expression by TRMs in the liver, spleen, lung and peritoneal cavity[27]. Prompted by these results, here we examine the role of RXRs in TRM development and maintenance in these tissues. RXR deficiency profoundly affects TRMs in the murine serous cavities. Lack of RXRs in myeloid cells impairs the neonatal expansion of peritoneal LPMs and leads to apoptotic cell death due to lipid accumulation. Furthermore, we show that peritoneal LPMs migrate into early ovarian tumours and that RXR deletion in myeloid cells diminishes peritoneal LPM accumulation in cancer lesions and reduces tumour progression in mice.

## Results

**RXR deficiency alters serous cavity macrophage populations.** ChIP-seq and ATAC-seq of TRMs purified from six organs revealed pronounced enrichment of RXRα in TRMs from liver, spleen, lung and peritoneum[27]. To assess whether RXRs control macrophage biology in these tissues, we deleted RXRα in myeloid cells by crossing Rxra^fl/fl mice[28] with mice expressing Cre in macrophages (LysM^Cre mice). RXRα deletion in LysM + cells caused a strong reduction of macrophage numbers in several tissues, including the liver, but the strongest impact was on peritoneal and pleural LPMs (Fig. 1a–c and Supplementary Fig. 1a–c). In contrast, circulating monocytes (Supplementary Fig. 1d) and other TRMs (Supplementary Fig. 1b) remained unaffected. In light of these results, we focused our study on serous macrophages.

There are three RXR isoforms: RXRα, RXRβ and RXRγ; however, peritoneal LPMs and SPMs expressed only Rxra and Rxrb transcripts (Supplementary Fig. 1e). We subsequently generated LysM^Cre+Rxra^fl/fl Rxrb^fl/fl double-knockout mice (LysM^Cre+Rxrab^fl/fl) to explore the role of these RXR isoforms in serous macrophage homeostasis. We observed a high LysM-mediated Cre recombination in peritoneal LPMs and SPMs from LysM^Cre+R26-YFP mice (Supplementary Fig. 1f), which correlated with an efficient reduction of both RXR isoforms in RXR-deficient LPMs and SPMs (Supplementary Fig. 1g). The reduction in serous LPM numbers was much stronger in mice lacking both RXRα and RXRβ in LysM cells than that observed in LysM^Cre +Rxra^fl/fl mice lacking only RXRα (Fig. 1d–f). Compared with wild-type controls, the serous cavities of both male and female LysM^Cre+Rxrab^fl/fl mice showed a marked reduction in embryo-derived TIM4^+ LPMs[16], whereas SPMs were significantly increased, and monocyte-derived TIM4^− LPMs remained unchanged (Fig. 1d–f and Supplementary Fig. 1h). The remaining serous LPM populations in RXR-deficient mice displayed an altered phenotype, with LysM^Cre+Rxra^fl/fl mice showing strong reductions in the expression of GATA-6 and CD102[11] (Supplementary Fig. 1i) and LysM^Cre+Rxrab^fl/fl mice showing increased expression of MHCII (Fig. 1f). These results establish the contribution of RXRα and RXRβ to the maintenance of the serous macrophage pool.

**RXRs control the identity of LPMs.** Gene expression profiling of RXR-deficient LPMs revealed major transcriptional changes (Benjamin–Hochberg adjusted p value ≤0.05) (Fig. 2a and Supplementary Tables 1 and 2). The most downregulated pathways relative to wild-type LPMs were the G₂/M checkpoint and the elongation factor 2 target modules, while the most enriched pathways included protein secretion, fatty-acid metabolism and apoptosis-related modules (Fig. 2b and Supplementary Fig. 2a). GATA-6, a retinoic acid-dependent transcription factor selectively expressed in LPMs[11,29], was reduced in RXR-deficient peritoneal LPMs (Supplementary Fig. 1i and 2b-d and Supplementary Table 1). We observed that 77 of the 184 genes downregulated in RXR-deficient LPMs were induced by retinoic acid treatment as recently reported[30] (Supplementary Fig. 2b), underscoring the importance of retinoic acid signalling in LPMs maintenance by RXRs. Those genes included peritoneal LPM hallmark genes, such as Gata6, Icam2 and Arg1, the cholesterol synthesis-related gene Cyp26a1, the regulator of proliferation Ccnd1 and the apoptosis inhibitors Naip1 and Cd5l

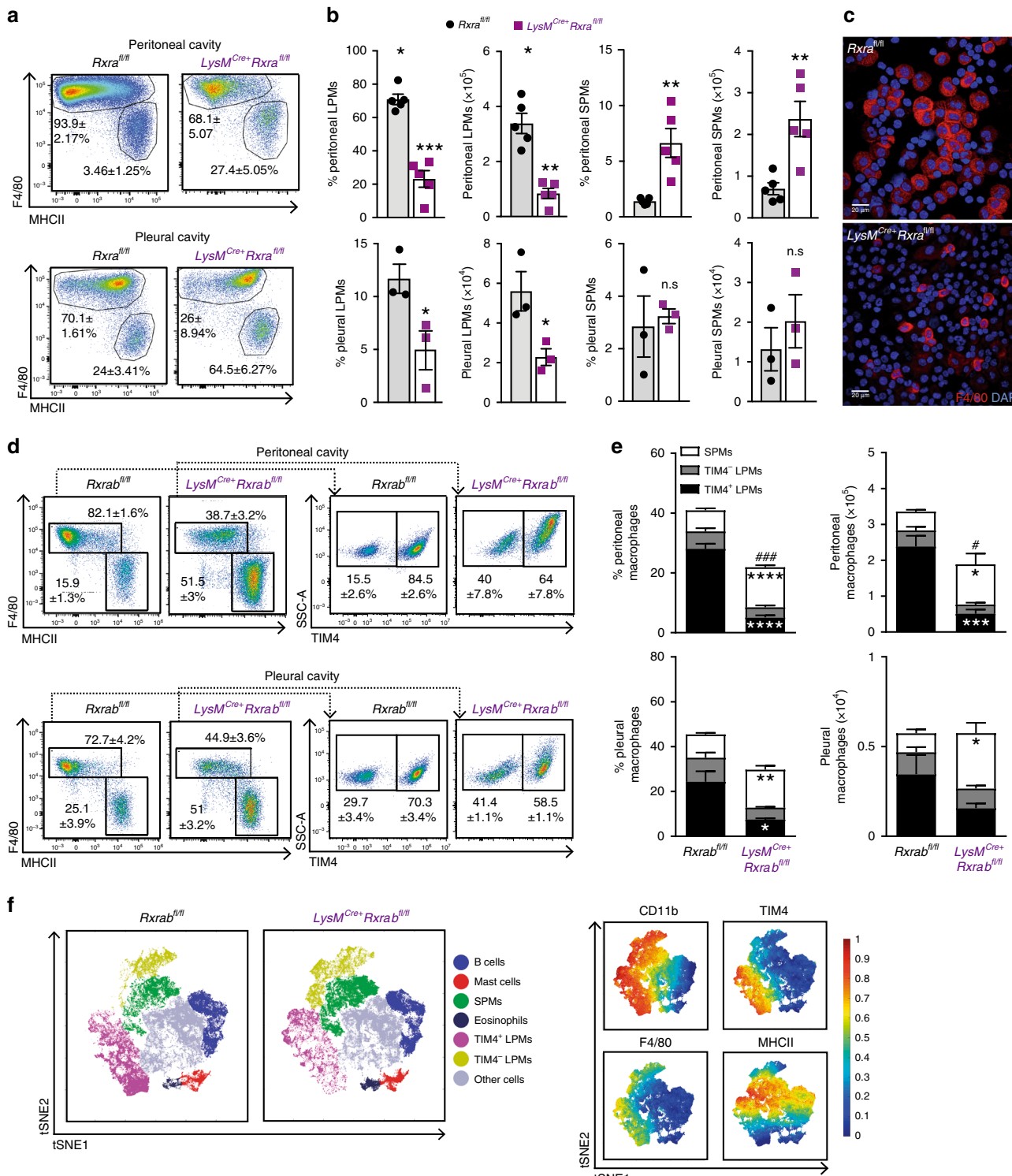

**Fig. 1 RXR deficiency alters serous cavity macrophage populations. a–b** Flow cytometry of serous cavities from 7 to 9-week-old *LysM*^Cre+*Rxra*^fl/fl mice and *Rxra*^fl/fl littermates. **a** Representative flow cytometry plots show the frequencies of LPMs (F4/80^HIMHCII^LO) and SPMs (F4/80^LOMHCII^HI) pregated on CD45 ^+B220^-CD11b^+CD115^+ cells (see also Supplementary Fig. 1a). **b** Graphs show frequencies among CD45^+ leukocytes and absolute numbers. *n* = 3–5 from at least two independent experiments. **c** Immunofluorescence images showing F4/80 and DAPI staining of peritoneal lavage cytospins from *LysM*^Cre+*Rxra*^fl/fl and *Rxra*^fl/fl mice. Scale bar: 20 µm. **d–e** Flow cytometry of serous cavities from 9-week-old *LysM*^Cre+*Rxrab*^fl/fl mice and *Rxrab*^fl/fl littermates (see also Supplementary Fig. 1h). **d** Representative flow cytometry plots show the frequencies of LPMs and SPMs pregated on CD45^+B220^-CD11b^+CD115^+ cells (left) and the percentages of TIM4^+ and TIM4^− cells pregated on LPMs (right). **e** Graphs show frequencies among CD45^+ leukocytes and absolute numbers. Data (*n* = 5–7 per genotype) are representative of two independent experiments; #*p* ≤ 0.05 (unpaired Student's *t* test) vs total macrophage percentage or absolute numbers in *Rxrab*^fl/fl mice; *\*p* ≤ 0.05; *\*\*p* ≤ 0.01; *\*\*\*p* ≤ 0.001; *\*\*\*\*p* ≤ 0.0001 (unpaired Student's *t* test) vs the same population in *Rxrab*^fl/fl mice. **f** Annotated t-SNE plots in the identified populations among CD45^+ peritoneal cells from *LysM*^Cre+*Rxrab*^fl/fl and *Rxrab*^fl/fl mice as in (**d**) (left) and overlaid with biexponential transformed marker expression levels (right). All data are presented as mean ± SEM; *\*p* ≤ 0.05; *\*\*p* ≤ 0.01; *\*\*\*p* ≤ 0.001 (unpaired Student's *t* test). Source data are provided as a Source Data file.

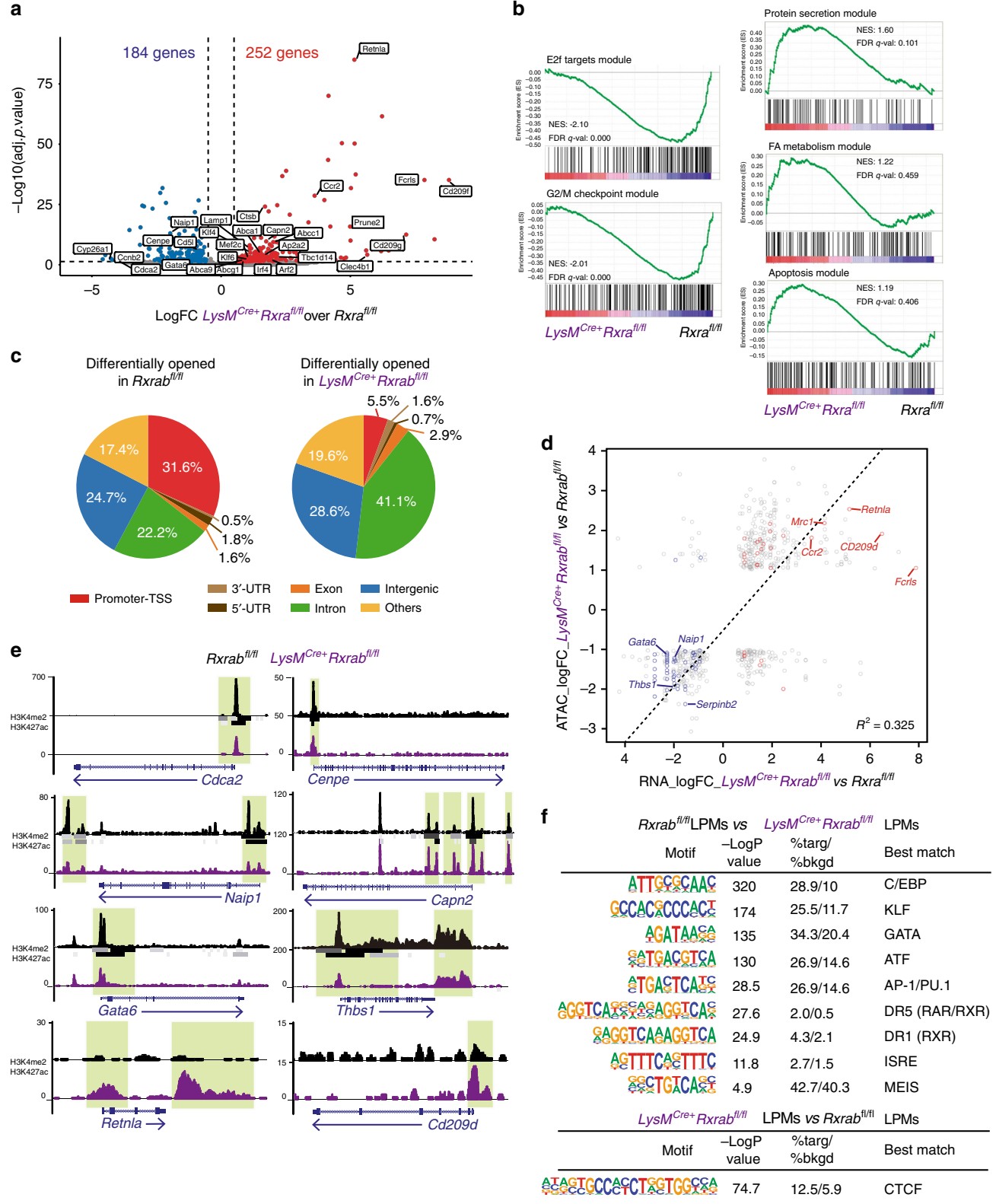

(Supplementary Fig. 2b). However, the expression profile of RXR-deficient peritoneal LPMs had limited overlap with the GATA-6-deficient macrophage expression profile[11,18,19] (Supplementary Fig. 2b–c), suggesting that RXRs control peritoneal LPMs via GATA6-dependent and -independent mechanisms. Some genes usually restricted to SPMs were upregulated in RXR-deficient LPMs, including *Irf4, Ccr2, Mrc1, Retnla, Fcrls, Cd209d* and

*Clec4b1*[16,20,31,32] (Fig. 2a, Supplementary Fig. 2c–d and Supplementary Table 2).

ATAC-seq revealed an altered distribution of chromatin-accessible peaks (Fig. 2c and Supplementary Fig. 2e) and a strong reduction of peaks in promoter-TSS (−1 kb to +200 bp with respect to TSS sites) in RXR-deficient LPMs compared with wild-type LPMs (5.5 vs 31.6%, Fig. 2c). Gene loci with increased

**Fig. 2 RXRs control the identity of LPMs. a** Volcano plot showing the global transcriptional changes in *LysM*<sup>Cre+</sup>*Rxra*<sup>fl/fl</sup> vs *Rxra*<sup>fl/fl</sup> LPMs determined by RNA-seq. Each circle represents one DEG and coloured circles represent DEGs significantly upregulated (Benjamini–Hochberg adjusted *p* value ≤0.05 and Log fold change (FC) ≥1.5 (in red)) or significantly downregulated (Benjamini–Hochberg adjusted *p* value ≤0.05 and Log FC ≤ 1.5 (in blue)). Normalized expression values from RNA-seq data are provided in Supplementary Tables 1 and 2. **b** Gene set enrichment analysis (GSEA) of RNA-seq data showing downregulated and upregulated functions in *LysM*<sup>Cre+</sup>*Rxra*<sup>fl/fl</sup> vs *Rxra*<sup>fl/fl</sup> LPMs. NES Normalized enrichment score; FDR false-discovery rate. **c** Genomic distribution of enriched regions in *LysM*<sup>Cre+</sup>*Rxrab*<sup>fl/fl</sup> and *Rxrab*<sup>fl/fl</sup> LPMs, identified in the ATAC-seq data set. **d** Scatter plot comparing accessibility to Tn5 transposase for differentially accessible peaks in *LysM*<sup>Cre+</sup>*Rxrab*<sup>fl/fl</sup> and *Rxrab*<sup>fl/fl</sup> LPMs (y axis, logFC in normalized read counts) and mRNA expression changes in *LysM*<sup>Cre+</sup>*Rxra*<sup>fl/fl</sup> and *Rxra*<sup>fl/fl</sup> LPMs for the nearest gene (x axis; logFC values). Grey dots represent the association between differentially accessible regions and the nearest differentially expressed genes, as detected by HOMER. Peaks associated with DEGs related to SPMs are highlighted in red (upregulated genes), and those related to the LPM-specific signature are highlighted in blue (downregulated genes). Chi-squared and Pearson correlation tests, $R^2 = 0.325$, Chi-square = 202.99, *p* value <10<sup>−6</sup>. **e** Genome browser views of proliferation-related (*Cdca2* and *Cenpe*), apoptosis-related (*Naip1* and *Capn2*), LPM-specific (*Gata6* and *Thbs1*) and SPM-specific (*Retnla* and *CD209d*) gene bodies in *LysM*<sup>Cre+</sup>*Rxrab*<sup>fl/fl</sup> and *Rxrab*<sup>fl/fl</sup> LPM ATAC-seq data set. H3K4me2 and H3K27ac-marked regions previously defined by Gosselin et al.[29] in LPMs from wild-type C57BL6/J mice are included as grey-to-black bars (tone intensity indicates read length). Vertical highlights correspond to regions of interest for the specified loci. **f** HOMER known motif analysis of *Rxrab*<sup>fl/fl</sup>- and *LysM*<sup>Cre+</sup>*Rxrab*<sup>fl/fl</sup>-specific ATAC-seq peak sequences in LPMs. Top table shows transcription factor motifs enriched in *Rxrab*<sup>fl/fl</sup> LPMs using a background corresponding to *LysM*<sup>Cre+</sup>*Rxrab*<sup>fl/fl</sup> LPM peaks. Bottom table shows transcription factor motifs enriched in *LysM*<sup>Cre+</sup>*Rxrab*<sup>fl/fl</sup> LPMs using a background corresponding to *Rxrab*<sup>fl/fl</sup> LPM peaks. Percentages of *Rxrab*<sup>fl/fl</sup> and *LysM*<sup>Cre+</sup>*Rxrab*<sup>fl/fl</sup> peaks relative to levels in their respective backgrounds are shown.

chromatin accessibility in RXR-deficient LPMs correlated with significantly upregulated gene expression (e.g. *Fcrls*, *Cd209d*, *Retnla*, *Mrc1* and *Ccr2*), while gene loci with reduced chromatin accessibility were associated with downregulated gene expression (e.g. *Gata6*, *Naip1*, *Thbs1*, *Serpinb2*) (Chi-squared and Pearson correlation tests, $R^2 = 0.325$; Chi-square = 202.99, *p* value <10<sup>−6</sup>) (Fig. 2d). The analysis of H3K4me2 and H3K427ac marks[29] showed active transcription of *Cdca*, *Cenpe*, *Naip1*, *Gata6* and *Thbs1* genes in wild-type peritoneal LPMs, and chromatin accessibility to these genes was significantly reduced in RXR-deficient peritoneal LPMs (Fig. 2e). Conversely, we observed increased regulatory element accessibility in genes upregulated in RXR-deficient peritoneal LPMs, such as *Capn2* (a positive regulator of apoptosis), *Retnla* and *Cd209d*[16,31] (Fig. 2e). RXR-deficient LPMs also showed a reduction in the number of peaks enriched in transcription factors required for LPM identity and maintenance (CEBP/B, KLF, GATA, ATF, ISRE, MEIS)[11,18,19,29,33], as well as for the nuclear receptors RXR (DR1) and RAR:RXR (DR5)[34] (Fig. 2f) (Motif enrichment analysis with HOMER using a hypergeometric test *p* value ≤ 0.01), suggesting direct binding of RXRs to specific genes. Conversely, binding sites for the insulator protein CTCF[35] were highly enriched in RXR-deficient LPMs (Fig. 2f), suggesting a potential repressor role as previously reported[36].

Our data indicate that RXRs are required to establish LPM identity through the regulation of chromatin accessibility and transcriptional regulation, and that this signature is in part dependent on retinoic acid signalling. Our data also suggest that RXR signalling might be important for lipid and protein-trafficking homeostasis in LPMs and for these cells' proliferation and survival.

**RXRs do not control the embryonic development of LPMs**. Since peritoneal LPMs arise from embryonic precursors that are recruited to the peritoneal cavity before birth, we asked whether RXRs control macrophage development in embryos. The analysis of previously published RNA-sequencing data sets[4] revealed that *Rxra* was mainly expressed in fetal liver macrophages from E10.5 to E18.5, whereas *Rxrb* was mainly expressed in earlier yolk sac and fetal liver progenitors (Supplementary Fig. 3a). Accordingly, qPCR analysis of purified yolk sac macrophages (CD11b<sup>INT</sup>Ly6C<sup>−</sup>F4/80<sup>HI</sup>), fetal liver macrophages (CD11b<sup>INT</sup>Ly6C<sup>−</sup>F4/80<sup>HI</sup>) and fetal liver monocytes (CD11b<sup>HI</sup>Ly6C<sup>+/−</sup>F4/80<sup>INT</sup>)[37] confirmed that *Rxra* was expressed in fetal liver macrophages, whereas *Rxrb* was expressed in yolk sac and fetal liver macrophages and monocytes

(Supplementary Fig. 3b–c). Absence of RXRs had no effect on yolk sac or fetal liver macrophages at E13.5, E15.5 or E18.5 or on peritoneal LPMs (CD11b<sup>+</sup>F4/80<sup>HI</sup>MHCII<sup>LO</sup>TIM4<sup>+</sup>) at E18.5 (Supplementary Fig. 3d). The recombination efficiency of *Rxra* and *Rxrb* alleles in *LysM*<sup>Cre+</sup>*Rxrab*<sup>fl/fl</sup> macrophage precursors was low before E18.5 (Supplementary Fig. 3e–f), and we therefore generated *Vav*<sup>Cre+</sup>*Rxrab*<sup>fl/fl</sup> mice, which show efficient Cre-mediated deletion of loxP-flanked alleles in hematopoietic cells during embryogenesis[38]. *Vav*<sup>Cre+</sup>*Rxrab*<sup>fl/fl</sup> compared with *Rxrab*<sup>fl/fl</sup> embryos showed no differences in the frequencies of yolk sac and fetal liver macrophages and peritoneal LPMs (Supplementary Fig. 3g), demonstrating that RXR signalling does not control the embryonic development of peritoneal LPMs.

**RXRs are required for neonatal expansion of LPMs**. We next examined whether RXRs control the maintenance of the peritoneal macrophage pool after birth. Most TRMs expand during the first week after birth and are maintained locally through prolonged survival and limited self-renewal[39]. We therefore analysed peritoneal lavage from *LysM*<sup>Cre+</sup>*Rxrab*<sup>fl/fl</sup> and *Rxrab*<sup>fl/fl</sup> mice from birth through postnatal day 70. Consistent with previous reports[16], almost all peritoneal macrophages in newborn mice expressed TIM4, and expanded significantly within the first 24 h of life and increased steadily with age (Fig. 3a). RXR-deficient peritoneal LPM population failed to undergo the neonatal expansion observed in wild-type littermates (Fig. 3a); moreover, this was accompanied by an increase in the number of peritoneal SPMs, which became the predominant phagocyte population in *LysM*<sup>Cre+</sup>*Rxrab*<sup>fl/fl</sup> mice by 42 days post-birth (Fig. 3b). Failure of RXR-deficient peritoneal LPMs to undergo the proliferation burst in the first days of life was confirmed by flow cytometry analysis of the nuclear proliferation antigen Ki-67 and 5-bromodeoxyuridine (BrdU) incorporation (Fig. 3c–d). This finding was consistent with the strong reduction in cell-cycle genes in RXR-deficient LPMs (Fig. 2a–b and Supplementary Fig. 2a). Validation of these results by qPCR analysis of purified peritoneal LPMs from *LysM*<sup>Cre+</sup>*Rxrab*<sup>fl/fl</sup> mice revealed upregulation of genes involved in the inhibition of cell-cycle progression (*Klf4* and *Klf6*) and downregulation of the G<sub>1</sub>/S checkpoint gene *Ccnd1* (cyclin D1) during the first day after birth (Fig. 3e). Interestingly, no alterations were observed in peritoneal LPMs of adult *Mx1*<sup>Cre+</sup>*Rxrab*<sup>fl/fl</sup> mice in which *Rxra* and *Rxrb* were conditionally deleted after the first week of life (Supplementary Fig. 4a–c). These results demonstrate an essential role

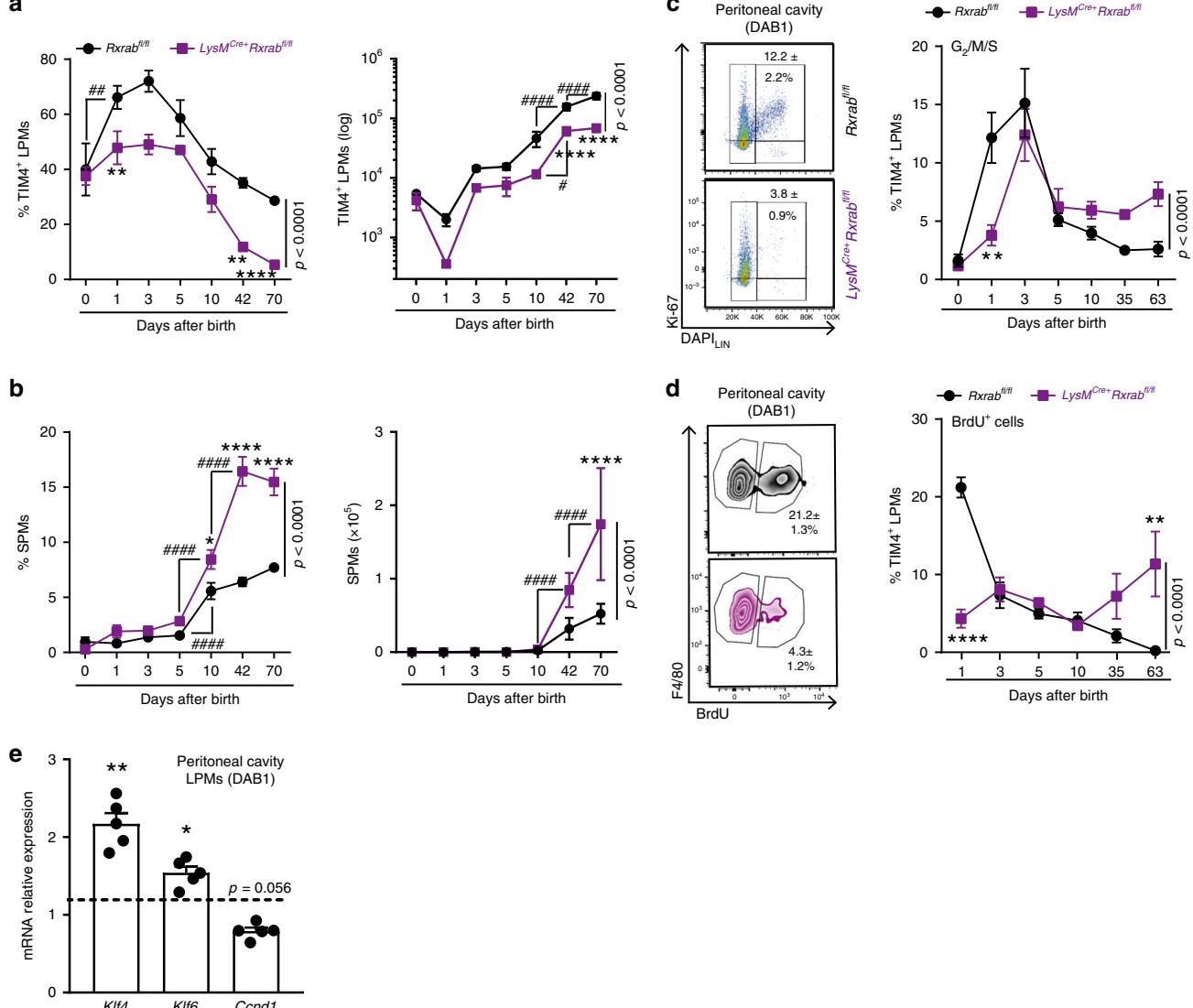

**Fig. 3 RXRs are required for the neonatal expansion of peritoneal LPMs. a–b** Frequency among CD45[+] cells and absolute numbers of TIM4[+] LPMs (**a**) and SPMs (**b**) from peritoneal exudates of *LysM*[Cre+]*Rxrab*[fl/fl] mice (purple) and *Rxrab*[fl/fl] mice (black) from the day of birth (0) through postnatal day 70. *n* = 3–17 per genotype and age, pooled from up to three independent experiments per age. **c** Representative dot plots showing Ki-67 and DAPI staining gated on peritoneal TIM4[+] LPMs, with quantification showing frequencies of proliferating (G₂/M/S) TIM4[+] LPMs over time. *n* = 4–9 per genotype and age, pooled from one to two independent experiments per age. **d** Flow cytometry density plots showing BrdU incorporation by TIM4[+] LPMs after treatment and quantification showing frequencies of TIM4[+] LPMs with BrdU incorporation. *n* = 3–12 per genotype and age, pooled from up to four independent experiments per age. **e** mRNA expression of cell-cycle-related genes in peritoneal LPMs from DAB1 *LysM*[Cre+]*Rxrab*[fl/fl] mice. Gene expression is normalized to DAB1 *Rxrab*[fl/fl] LPMs (dashed line). *n* = 5 per genotype and gene. All data are presented as mean ± SEM; *$p \leq 0.05$, **$p \leq 0.01$, ***$p \leq 0.001$ and ****$p \leq 0.001$ compared with age-paired *Rxrab*[fl/fl] mice; #$p \leq 0.05$, ##$p \leq 0.01$, ###$p \leq 0.001$ and ####$p \leq 0.001$; (**a–d**) two-way ANOVA followed by Tukey's multiple comparisons test; and (**e**) unpaired Student's *t* test. DAB: day after birth. Source data are provided as a Source Data file.

for RXRs in the regulation of postnatal expansion of peritoneal LPMs.

**RXR deficiency leads to lipid accumulation and apoptosis of adult LPMs.** Oil-red staining revealed a higher abundance of lipid-filled vacuoles in RXR-deficient peritoneal TIM4[+] LPMs than in wild-type TIM4[+] LPMs (Fig. 4a). Increased lipid accumulation was not due to increased uptake, since RXR-deficient peritoneal LPMs downregulated genes involved in cholesterol uptake (*Ldlr*, *Fabp4*, *Fabp5* and *Msr1*) and synthesis (*Cyp26a1*) (Supplementary Fig. 4d) and upregulated genes involved in cholesterol efflux (*Abcc1*, *Abca1*, *Abca9* and *Abcg1*) and lipid metabolism (*Lpl*) (Supplementary Fig. 4d). The upregulated genes

in RXR-deficient peritoneal LPMs also included an enriched profile of lysosomal-related transcripts, including structural genes (*Lamp1*), hydrolases (*Ctsb*, *Ctsc*, *Ctsl* and *Naaa*), protein pumps required for lysosome acidification (*Atp6ap1*, *Atp6v1c1*, *Atp6ap2* and *Ap2m1*), molecules involved in phagosome maturation (*Tlr4*, *Rab11fip5* and *Rab11a*), sortin nexins implicated in lysosome transport of apoptotic cells[40] (*Snx2*, *Snx5*, *Snx6* and *Snx13*), and the negative autophagy regulator *Tbc1d14*[41] (Fig. 4b). Consistent with these findings, staining with LysoTracker (LT), a fluorescent weak amine that accumulates in lysosomes and autolysosomes, detected highly abundant acidic vesicular organelles in TIM4[+] LPMs from *LysM*[Cre+]*Rxrab*[fl/fl] mice (Supplementary Fig. 4e). Imaging of sorted peritoneal macrophages revealed two classes of acidic lysosomes in RXR-deficient TIM4[+] LPMs: presumably

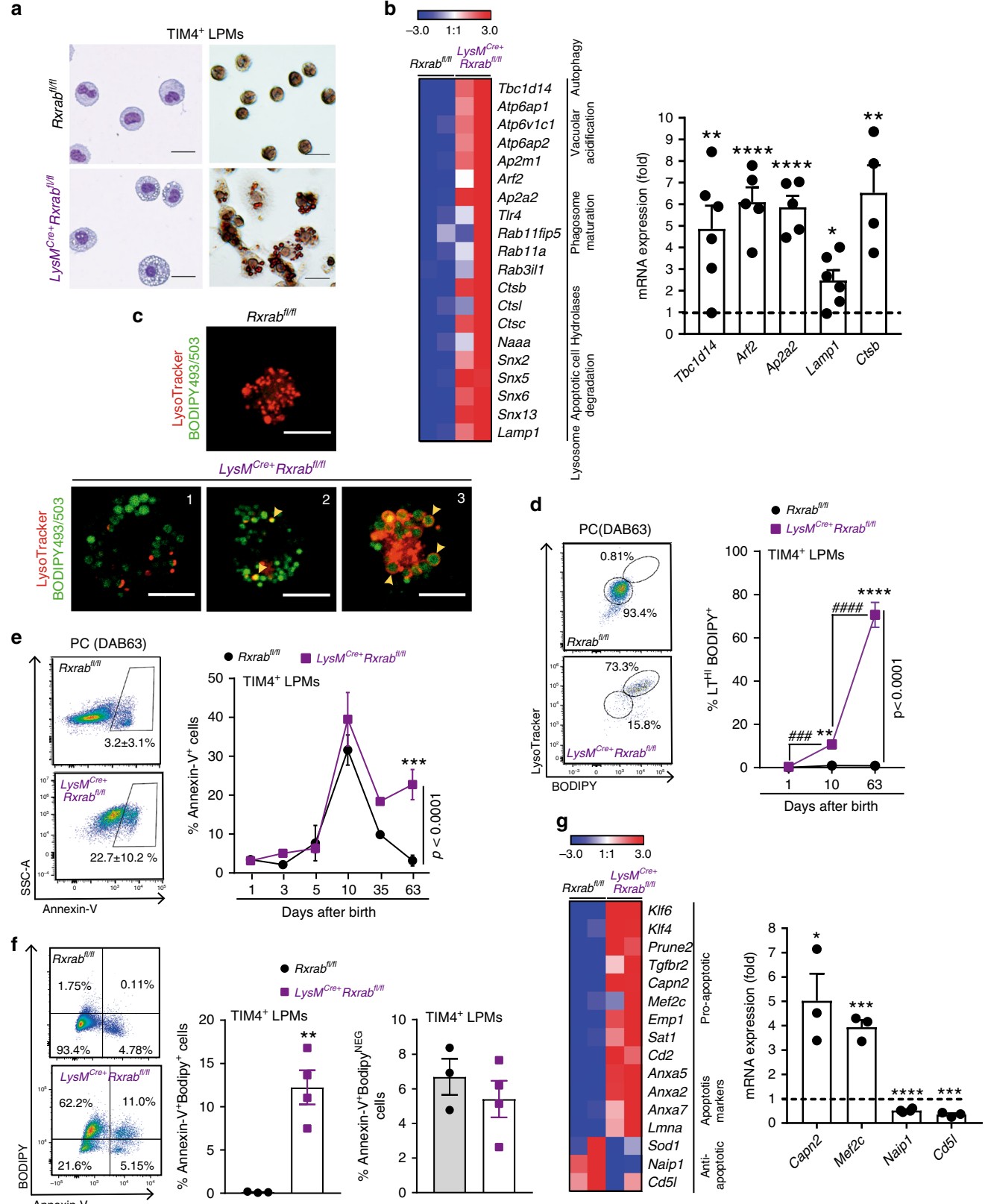

lipophagic vesicles staining positively for BODIPY and LT (Fig. 4c, panel 2 and Supplementary Fig. 4f) and larger vesicles with an acidic ring surrounding a central core of neutral lipids (Fig. 4c, panel 3 and Supplementary Fig. 4f), likely derived from phagocytized cell bodies[42]. Furthermore, flow cytometry analysis of cells from neonatal and adult *LysM*^Cre+^*Rxrab*^fl/fl^ mice revealed

a progressive accumulation of LT-BD-double-stained peritoneal TIM4+ LPMs with age (Fig. 4d).

The lipid cargo of macrophages influences their phenotype, and excessive lipid loading has been linked to impaired cell survival, often caused by endoplasmic reticulum stress and lysosome dysfunction[43,44]. Consistent with this observation,

**Fig. 4 RXR deficiency leads to lipid accumulation and apoptosis of peritoneal TIM4$^+$ LPMs in adult mice. a** Representative May–Grünwald–Giemsa cytospins of sorted TIM4$^+$ LPMs (left), and Oil Red O stained TIM4$^+$ LPMs (right) from 9-week-old *LysM*$^{Cre+}$*Rxrab*$^{fl/fl}$ and *Rxrab*$^{fl/fl}$ mice. Scale bars: 20 μm (May–Grünwald–Giemsa) and 10 μm (Oil Red O). **b** Heatmap showing normalized log2 FC in the expression of protein transport-related genes in adult *LysM*$^{Cre+}$*Rxra*$^{fl/fl}$ and *Rxra*$^{fl/fl}$ LPMs (left), and qPCR analysis of *LysM*$^{Cre+}$*Rxrab*$^{fl/fl}$ and *Rxrab*$^{fl/fl}$ LPMs (right). $n = 4$–6 per genotype and gene. **c** Confocal images showing overlaid channels for LysoTracker (red) and BODIPY493/503 (green) from sorted and cultured TIM4$^+$ LPMs from adult *LysM*$^{Cre+}$*Rxrab*$^{fl/fl}$ and *Rxrab*$^{fl/fl}$ mice (see also Supplementary Fig. 4f). Panels show a representative image of *Rxrab*$^{fl/fl}$ LPMs and three representative fields of view of *LysM*$^{Cre+}$*Rxrab*$^{fl/fl}$ TIM4$^+$ LPMs: (1) cells with non-acidic lipid vesicles; (2) cells with lipid and lysosome markers overlapped (yellow arrowhead); and (3) cells with a central lipid core surrounded by an acidic ring (yellow arrowheads). Scale bar = 10 μm. **d** Flow cytometry dot plots showing BODIPY493/503 and LysoTracker (LT) staining gated on TIM4$^+$ LPMs (left), and the percentage of double-positive LT$^{HI}$BODIPY$^+$ TIM4$^+$ LPMs from *LysM*$^{Cre+}$*Rxrab*$^{fl/fl}$ and *Rxrab*$^{fl/fl}$ mice (right). $n = 4$–8 per genotype and age, pooled from one to two independent experiments per age. **e** Flow cytometry dot plots showing Annexin-V staining gated on TIM4$^+$ LPMs (left), and frequency quantification of apoptotic TIM4$^+$ LPMs (right) in *LysM*$^{Cre+}$*Rxrab*$^{fl/fl}$ and *Rxrab*$^{fl/fl}$ mice. $n = 3$–9 per genotype and age, pooled from up to three independent experiments per age. **f** Flow cytometry dot plots showing Annexin-V and BODIPY493/503 staining gated on TIM4$^+$ LPMs (left), and frequency quantification of lipid-loaded and lipid-free apoptotic TIM4$^+$ LPMs (right) in adult *LysM*$^{Cre+}$*Rxrab*$^{fl/fl}$ and *Rxrab*$^{fl/fl}$ mice. $n = 3$–4 per genotype, representative of four independent experiments. **g** Heatmap showing normalized log2 FC in the expression of apoptosis-associated genes in adult *LysM*$^{Cre+}$*Rxra*$^{fl/fl}$ and *Rxra*$^{fl/fl}$ LPMs (left), and qPCR analysis of *LysM*$^{Cre+}$*Rxrab*$^{fl/fl}$ and *Rxrab*$^{fl/fl}$ LPMs (right). $n = 3$–4 per genotype and gene. qPCR data are presented as gene expression in *LysM*$^{Cre+}$*Rxrab*$^{fl/fl}$ normalized to *Rxrab*$^{fl/fl}$ LPMs (dashed line). All data are presented as mean ± SEM; *$p \leq 0.05$; **$p \leq 0.01$; ***$p \leq 0.001$; ****$p \leq 0.0001$; (**d** and **e**) two-way ANOVA and (**b**, **f** and **g**) unpaired Student's *t* test. Source data are provided as a Source Data file.

Annexin-V$^+$ cells were significantly increased in lipid-loaded cells of RXR-deficient TIM4$^+$ LPMs from adult mice (Fig. 4e–f). Apoptosis induction correlated with transcript upregulation of pro-apoptotic genes (*Capn2* and *Mef2c*) and downregulation of the antiapoptotic gene mRNA programme (*Naip-1* and *Cd5l*) in RXR-deficient peritoneal LPMs from adult mice (Fig. 4g). Together, these results indicate that absence of RXRs leads to progressive lipid accumulation in serous TIM4$^+$ LPMs, resulting in LPM death through apoptosis.

**Peritoneal macrophages promote ovarian cancer progression.** Ovarian cancer is the most lethal gynaecological disease in women, with ~300,000 new diagnoses worldwide in 2018 and an overall survival rate of <40%[45]. While the prognosis for most solid tumours has improved, treatment of epithelial ovarian cancer has advanced little, remaining based on surgery, hormone therapy and chemotherapy, with only one major new treatment introduced in the last 30 years[46]. An in-depth understanding of the cellular and molecular mechanisms of ovarian cancer progression are crucial to overcoming this life-threatening disease.

Intracelomic migration of serous macrophages into solid organs has been shown to contribute to wound healing and tissue remodelling[47]. Since tissue remodelling can also contribute to cancer progression, we examined whether peritoneal LPMs play a role in the early stages of ovarian cancer progression. We induced ovarian cancer in an orthotopic syngeneic model by injecting Upk10 ovarian cancer cells into the ovarian bursa of immunocompetent C57BL/6 mice[48] (Supplementary Fig. 5a). Upk10 cells derive from advanced murine ovarian tumours with concurrent ablation of p53 and activation of oncogenic K-ras[48]. As previously described, this tumour model recapitulates the immune populations and cytokine milieu of human tumours[48]. Primary ovarian solid tumours were palpable 3 weeks post-Upk10 injection (Supplementary Fig. 5b). Flow cytometry analysis of tumour-infiltrating leukocytes within the primary ovarian tumour revealed a progressive accumulation of CD45+ leukocytes (Supplementary Fig. 5c). The main myeloid cell population that accumulate in early ovarian tumours (2–3 weeks) were CD11b$^+$F4/80$^+$ macrophages (Supplementary Fig. 5c). Ovarian tumours in wild-type mice had an elevated content of F4/80$^+$GATA-6$^+$ and F4/80$^+$GATA-6$^-$ macrophages, whereas healthy ovaries contained only F4/80$^+$GATA-6$^-$ macrophages (Supplementary Fig. 6a), a finding confirmed by confocal microscopy (Supplementary Fig. 6b). The expression of GATA-6 in F4/80$^+$

macrophages strongly supported their peritoneal identity, which was further confirmed by their expression of the peritoneal LPM-specific marker CD102 (Supplementary Fig. 6c). We analysed primary ovarian tumours in *LysM*$^{Cre+}$*Rxrab*$^{fl/fl}$ and *LysM*$^{Cre+}$*Rxrab*$^{fl/fl}$ mice 24 days after Upk10 intrabursal inoculation (Fig. 5a). Strikingly, ovarian tumour progression was markedly delayed in *LysM*$^{Cre+}$*Rxrab*$^{fl/fl}$ mice compared with their control littermates (Fig. 5a–c). Tumour growth delay in *LysM*$^{Cre+}$*Rxrab*$^{fl/fl}$ mice was associated with a marked reduction in the numbers of tumour-infiltrating F4/80$^+$GATA-6$^+$ macrophages (Fig. 5d–f and Supplementary Fig. 6d), whereas no differences from control littermates were observed neither in the frequency of tumour-infiltrating F4/80$^+$GATA-6$^-$ macrophages (Supplementary Fig. 6e) nor in macrophages within ovaries from naïve *LysM*$^{Cre+}$*Rxrab*$^{fl/fl}$ mice (Supplementary Fig. 6f). Neutrophils and monocytes were unlike to contribute to the phenotype observed in *LysM*$^{Cre+}$*Rxrab*$^{fl/fl}$ mice due to the lack of expression of RXRα in these populations[27,49]. However, since monocyte-derived macrophages express RXRα (Supplementary Fig. 6g-h), a functional role for these cells cannot be excluded. These results provide the first evidence that peritoneal LPMs infiltrate ovarian tumours and contribute to tumour progression, and further suggest that immunomodulation of RXR-dependent peritoneal LPMs offers a potential therapeutic strategy for ovarian cancer.

## Discussion

Although TRMs are known to persist in tissues for prolonged periods, the mechanisms that promote the homeostasis and maintenance of these cells remain unclear. Here, we found that the nuclear receptors RXRα and RXRβ control LPMs in both the peritoneal and pleural cavities. For technical reasons, our analysis focused principally on RXR-dependent LPM regulation in the peritoneal cavity. We demonstrated that RXRα and RXRβ play a key role in the expansion and maintenance of LPMs through their ability to control LPM proliferation in the postnatal period and LPM lipid metabolism and survival during adulthood. We also demonstrate for the first time that RXR-dependent peritoneal LPMs infiltrate early ovarian tumours and contribute to tumour progression.

RXRs can act as self-sufficient homodimers but also form obligate heterodimers with other nuclear receptors, including RARs (retinoic acid receptors)[23]. Using ATAC-seq profiling, we identified here an enrichment in chromatin motifs known to bind both RAR/RXR and RXR homodimers (DR5 and DR1 elements,

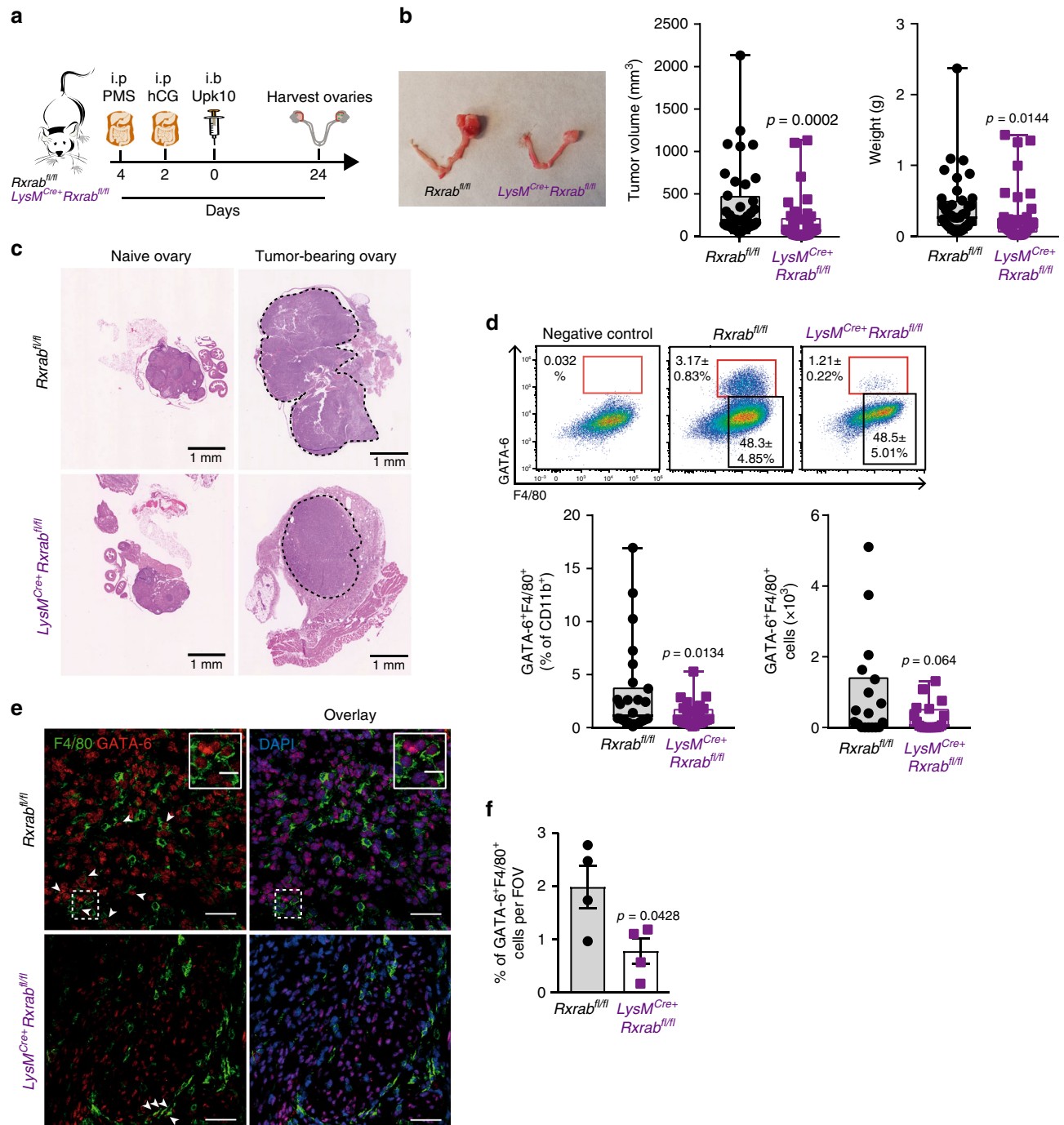

**Fig. 5 Peritoneal macrophages invade early ovarian tumour lesions and promote ovarian cancer progression. a** Experimental design of the orthotopic model for Upk10 ovarian tumours. i.p. intraperitoneal; i.b. intrabursal; PMS pregnant mare serum gonadotropin; hCG human chorionic gonadotropin; Upk10 mouse ovarian tumour cell line. **b** Representative image of ovarian tumours (left) and quantification of primary ovarian tumour size and weight (right) in *Rxrab*[fl/fl] and *LysM*[Cre+]*Rxrab*[fl/fl] mice. Data are presented as median (lower-upper quartiles and minimum and maximum values). $n = 37$–39 per genotype, pooled from seven independent experiments. **c** Hematoxylin and eosin staining of naïve and tumour-bearing ovaries in the same mice as in (**b**). Scale bar = 1 mm. Dotted lines delimit the tumour border. **d** Flow cytometry analysis of GATA-6[+]F4/80[+] macrophages (red box gate) in primary ovarian tumours (top). Frequency of GATA-6[+]F4/80[+] macrophages gated as CD45[+]Upk10[-]B220[-]Ly6G[-]CD3[-]CD11b[+] cells (bottom, see also Supplementary Fig. 6d). Negative control shows non-specific staining with the secondary antibody for GATA-6. Data are presented as median (lower-upper quartiles and min-max values). $n = 18$–27 per genotype, pooled from four independent experiments. **e** Immunofluorescence staining for GATA-6 (red), F4/80 (green) and DAPI (blue) on frozen tumour sections from the same mice as in (**b**). Scale bars: 100 and 10 μm for insets (white squares). Data are representative of four animals per genotype. **f** Quantification of GATA-6[+] F4/80[+] cells in the images in (**e**). Data are presented as mean ± SEM. $n = 4$ per genotype. *$p \leq 0.05$; ****$p \leq 0.0001$; (**b**) Mann–Whitney $U$ test, (**d**) generalised lineal model with gamma distribution and (**f**) unpaired Student's $t$ test. Source data are provided as a Source Data file.

respectively)[23] in WT but not RXR-deficient peritoneal LPMs, suggesting direct binding of RXRs to specific genes. Wt1+ stroma have been recently demonstrated to secrete retinoic acid metabolites that sustains LPM identity[29,30]. In our studies, we show that a large number of genes downregulated upon RXR lost are dependent on retinoic acid signalling. These findings suggest that RXRs control retinoic acid-mediated transcriptional regulation of LPMs through heterodimerization with RAR and that other RXR heterodimers and/or RXR homodimers might control specific functional programmes of LPMs, either collaboratively or independently.

TRMs develop in the embryo, undergo a burst of proliferation post birth and survive locally for prolonged periods during adulthood through slow self-renewal[39,50]. Postnatal TRM survival in most tissues is controlled by CSF-1 and its receptor (CSF-1R)[51–54]. However, the signals that regulate proliferation at neonatal stages are unknown. We demonstrate here that RXRs do not control the differentiation of peritoneal LPMs in embryos, but that the absence of RXRs dramatically impairs postnatal LPM expansion, resulting in a severely reduced LPM pool in the serous cavities. Our data support the relevance of RXR-dependent local proliferation during the first hours of life, since perinatal impaired proliferation is sufficient to reduce the pool of embryo-derived peritoneal macrophages throughout the life of $LysM^{Cre+}Rxra^{fl/fl}$ mice. Supporting this conclusion, adult $Mx1Cre^{Cre+}Rxrab^{fl/fl}$ mice showed no alterations in peritoneal cavity LPM content. Although we found that peritoneal LPMs from adult $LysM^{Cre+}Rxrab^{fl/fl}$ mice had an elevated proliferation rate, LPMs remained absent from adult serous cavities. Our results show that RXR deficiency increases lipotoxicity and susceptibility to apoptotic cell death in peritoneal LPMs expressing the phagocytic receptor TIM4[55], suggesting that lipid accumulation in these cells might be linked to defective processing of apoptotic cargo[56,57].

Ovarian cancer is one of the most common gynaecologic cancers that has the highest mortality rate[45]. There are a number of ongoing clinical trials in ovarian cancer, including PARP inhibitors, which target tumours' DNA repair capabilities[58]. However, new strategies are needed to improve ovarian cancer care and patient survival. Interestingly, a recent study has demonstrated that peritoneal-resident macrophages facilitate tumour progression through their ability to increase itaconate production in cancer cells[59]. Our finding that peritoneal LPMs infiltrate early mouse ovarian cancer lesions indicate a resemblance to tumour-associated macrophages (TAMs) described in human ovarian tumours, which express high levels of mouse LPM-specific markers, including GATA-6, ADGRE1 (F4/80), TIMD4 and RXRα[60]. These results are in line with a recent study showing that peritoneal resident macrophages transmigrate into injured livers to promote tissue repair[47], demonstrating that peritoneal resident macrophages are not stationary and can rapidly move to neighbouring tissues in pathological or premalignant conditions. The markedly reduced ovarian tumour progression in mice with myeloid RXR deletion has important implications for future clinical strategies in ovarian cancer. In summary, our results identify RXRs as key regulators of serous macrophage expansion in the neonatal phase and their subsequent maintenance, and suggests that modulation of RXR-dependent LPMs might delay ovarian cancer progression.

## Methods
**Mice**. All the animals used in this study were on the C57BL/6 background. Mice were housed at 2–5 animals per cage with a 12-h light/dark cycle (lights on from 07:00 to 19:00 hrs) at constant temperature (23 °C) with ad libitum access to food and water. $LysM^{Cre}Rxra^{fl/fl}$[24] and $Mx1^{Cre}Rxra^{fl/fl}Rxrb^{fl/fl}$[25] mice were previously generated. To conditionally ablate $Rxra$ and $Rxrb$ we crossed $LysM^{Cre}$ or $Vav^{Cre}$[38] mice with mice carrying $Rxra^{fl/fl}$ and $Rxrb^{fl/fl}$[28,61]. The heterozygous offspring containing lox-P-targeted $Rxra$ ($Rxra^{fl/+}$) and $Rxrb$ ($Rxrb^{fl/+}$) genes, and the Cre

transgene ($LysM^{Cre}Rxra^{fl/+}+Rxrb^{fl/+}$ or $Vav^{Cre}Rxra^{fl/+}+Rxrb^{fl/+}$) were then crossed with $Rxra^{fl/fl}Rxrb^{fl/fl}$ mice to generate mice homozygous for the $Rxra$ and $Rxrb$ floxed alleles: $LysM^{Cre+}Rxra^{fl/fl}Rxrb^{fl/fl}$ ($LysM^{Cre+}Rxrab^{fl/fl}$) or $Vav^{Cre+}Rxra^{fl/fl}Rxrb^{fl/fl}$ ($Vav^{Cre+}Rxrab^{fl/fl}$). Their $LysM^{Cre-}Rxra^{fl/fl}Rxrb^{fl/fl}$ or $Vav^{Cre-}Rxra^{fl/fl}Rxrb^{fl/fl}$ littermates were used as controls ($Rxrab^{fl/fl}$). Mice were genotyped by PCR using the following primers: P1 and P2 for $Rxra$ (800 pb); XO141 and WS55 for $Rxrb$ (270 pb)[25]; Cre1 and Cre2 for Cre (450 pb). Primer sequences are provided in Supplementary Table 5. To assess the efficiency and specificity of Cre-mediated recombination in macrophages we crossed the $LysM^{Cre}$ strain to ROSA26-flox-stop-flox-EYFP reporter (ROSA-EYFP) mice[62]. Male and female mice were studied from E13.5 to 70 days of age. In studies using DAB0 pups, 2 mg progesterone (Sigma) was i.p. injected to pregnant females at E17.5 and E18.5. At 19.5, pregnant mice were euthanized and fetuses were obtained by C-section. Before tissue harvest, adult animals were killed by carbon dioxide ($CO_2$) asphyxiation, pregnant females by cervical dislocation and embryos and neonates by decapitation. All experiments were performed according to local ethical guidelines and were approved by the Animal Subjects Committee of the Instituto de Salud Carlos III (Madrid, Spain) in accordance with EU Directive 86/609/EEC, or by IACUC at Icahn School of Medicine at Mount Sinai in accordance with NIH guidelines.

**Cell isolation**. Cell suspensions were enriched from peritoneal and pleural lavage, lung, liver, small intestine, brain, spleen, ovaries, yolk sac and fetal liver. For the ovarian tumour model, we induced estrus synchronization by i.p. injection of 5U pregnant mare serum gonadotropin (PMSG) and 5U of human chorionic gonadotropin (hGC) 96 and 48 h prior to Upk10 injection, respectively. Peritoneal and pleural cavities from adult mice were washed with 10 and 2 mL of sterile PBS 1X, respectively. In E18.5 embryos and DAB1-10 neonates, the peritoneal cavity was washed with 1 mL of sterile PBS 1× and recovered by massaging the abdominal cavity surface. Yolk sac and fetal liver were chopped finely and digested with 1 mL of 0.1 mg/mL collagenase type IV (Sigma) at 37 °C for 30 min (yolk sac) or 5 min (fetal liver). For lungs, brain, spleen, liver and ovaries, mice were perfused with 20 mL of cold sterile PBS 1X through the left ventricle. Lungs were digested with 0.8 mg/mL collagenase type IV (Sigma) at 37 °C for 30 min. Brains were triturated using a scalpel and then digested with 1 mL of 4 mg/mL papaine (Worthington) at 37 °C for 30 min. Spleens were smashed using a 100 μM nylon filter. Livers were digested with 0.8 mg/mL of liberase (Roche) at 37 °C for 30 min. Digested livers were centrifuged at $50 \times g$ for 3 min and supernatants were resuspended in 40% Percoll and centrifuged at $800 \times g$ for 20 min without acceleration nor break. Ovaries were chopped finely and digested with 1 mL of 0.1 mg/mL collagenase type IV (Sigma) at 37 °C for 30 min. DNase I (Sigma) was added to all the digestions at 0.2 mg/mL. All tissues were passed through a 100 μM nylon filter and in those tissues with remaining blood depots, red blood cells were lysed using 1× RBC Lysis Buffer (eBioscience).

**Flow cytometry and cell sorting**. Cell suspensions were prepared as described above. Cells (up to $5 \times 10^6$) were blocked using anti-CD16/32 (BioLegend) and thereafter stained with appropriate antibodies. A full list of antibodies and gating strategies are provided in Supplementary Tables 3 and 4, respectively. To measure Ki-67 profile cells were fixed and permeabilized according to the Foxp3 staining buffer set (ThermoScientific) and incubated with a Ki-67 antibody (Biolegend) and DAPI. To measure BrdU incorporation in proliferating cells, BrdU (BD Biosciences) was i.p. injected in mice: 1 mg of BrdU (adult mice) or 0.1 mg (for neonates). Two hours later peritoneal lavages were obtained and cells were fixed and permeabilized. BrdU staining was performed following the FITC BrdU flow kit instruction manual (BD Biosciences). To study apoptosis, cells were labelled with Annexin-V antibody (Biolegend) in Annexin-V buffer (BD Biosciences). After washing, cells were stained with DAPI and analysed by FACS within 1 h after the labelling. For Lysotracker and Bodipy staining, unfixed cells were incubated in RPMI media containing 10% fetal bovine serum (FBS), 1% antibiotics (Pen/Strep) and 70 nM LysoTracker (Invitrogen) in the dark at 37 °C for 1 h. Cells were washed twice with PBS 1×. Cells were incubated with 1:2000 BODIPY493/503 (Thermo) for 20 min and washed twice with PBS 1×. Flow cytometry was performed using Fortessa (BD, Biosciences), Canto 3 L (BD, Biosciences) or SP6800 Spectral Analyzer (SONY) and data were analysed using FlowJo 10.4.2 Software. For sorting, we used an ARIA SORT (BD, Biosciences) or a customised SY3200 Cell Sorting (SONY).

**t-SNE analysis**. FCS files from four $LysM^{Cre+}Rxrab^{fl/fl}$ and five $Rxrab^{fl/fl}$ mice were processed using AP-workflow[63] an automated pipeline for the analysis and visualization of high-dimensional flow cytometry data. The specified surface markers included in the staining panel (CD45, B220, CD11b, F4/80, MHCII and TIM4) were used for t-Distributed Stochastic Neighbour Embedding (t-SNE) dimensionality reduction after automatic biexponential transformation of compensated channels. Leukocytes from $LysM^{Cre+}Rxrab^{fl/fl}$ and $Rxrab^{fl/fl}$ samples were processed together (≈1 million). Subpopulations were manually gated from t-SNE maps with FlowJo software (FlowJo, Ashland, OR). MATLAB R2017b (The MathWorks Inc., Natick, MA) was used to render t-SNE maps overlaid with marker expression levels. t-SNE maps coloured by subpopulation were depicted for

*LysM*^Cre+^*Rxrab*^fl/fl^ and *Rxrab*^fl/fl^ mice separately using transparency to include density information within the representation.

**Quantitative real time PCR (Q-PCR)**. Total RNA was isolated using Trizol (Sigma) and Max Extract High Density tubes (Qiagen). Transcripts were quantified using the system AB7900-FAST-384 with a two-step reverse-transcription qPCR process. Gene expression values were normalized to housekeeping genes *36b4* and *cyclophilin* and expressed as relative mRNA levels or fold changes compared with littermate controls. Data were analysed using qBASE (Biogazelle). Primer sequences are provided in Supplementary Table 5.

**RNA-sequencing processing and analysis**. LPMs were double-sorted in phenol-red free DMEM, 0.1% azide, 10 mM HEPES and 2% FCS, which resulted in a 99% pure LPM population. Cells were lysed in RLT buffer and RNA was extracted according to the protocol provided by the RNeasy Mini Kit (Qiagen, USA). cDNA synthesis and library preparation were performed using the SMART-Seq v4 Ultra Low Input RNA Kit and Low Input Library Prep Kit v2 (Clontech), respectively. Sequencing was performed using the Illumina NextSeq-500 system. Transcript abundances were quantified with the Ensembl GRCh38 cDNA reference using Kallisto version 0.43.0. Transcript abundances were summarized to gene level using tximport. The expression matrix was filtered for only transcripts with greater than 5 TPM in replicates. Only genes with at least 1 count per million in the two replicates were considered for statistical analysis. Differential expression statistics between different macrophage subsets were generated using limma with TMM normalization. Genes with a Benjamini–Hochberg adjusted $p$ value <0.05 and a cut-off of 0.6 in logFC were considered differentially expressed (DEGs). GO term enrichment was performed using PANTHER (ontology: biological process) and Gene Set Enrichment Analysis (GSEA) software. For PANTHER analysis Bonferroni correction was applied. Heatmaps were built and normalized using Genesis 1.7.6.

**Oil red O staining**. Cultured cells were washed twice with PBS 1× and fixed with paraformaldehyde 4%. Cells were washed with isopropanol 60% and let dry. Preparations were incubated with oil red for 15 min at RT and washed with distilled water. Finally, slides were incubated for 30 s in hematoxylin and washed with distilled water. Images were acquired using an Olympus microscope with X40 magnification objective and analysed with Fiji Imaje J Software.

**Chromatin profiling by assay for transposase-accessible chromatin with sequencing (ATAC-Seq)**. Epigenomic profiling of chromatin accessibility was assessed by ATAC-seq as described by Buenrostro al[64]. Briefly, 40,000 sorted peritoneal LPMs per replicate were centrifuged at $500 \times g$ for 20 min at 4 °C. Cells were lysed with lysis buffers (10 mM Tris-HCl pH 7.4, 10 mM MgCl2, 0.1% IGEPAL CA-630). Transposase reaction was performed incubating the isolated nuclei with 2.5 mL per sample of Tn5 from the Nextera DNA Library Preparation Kit (Illumina) for 30 min at 37 °C. Transposed DNA was purified with the ChIP DNA Clean & Concentrator kit (Zymo) according to manufacturer's instructions. PCR amplification and barcoding were done with the primers described in Supplementary Table 5. Each PCR reaction included 11 μL NEB 2 × PCR Mix (New England Biolabs), 10 μL of transposed DNA, 0.5 μL of primer Ad_ no Mx (forward) and 1 × 0.5 μL of of barcoded reverse primer (Ad_2.1 to Ad_2.4). PCR conditions were as follows: 72 °C for 5 min, 98 °C for 30 s, 5 cycles of 98 °C for 10 s, 63 °C for 30 s, 72 °C for 1 min, followed by 4 °C next to final 5th cycle. After first PCR DNA was size selected with 0.5X volume SPRI beads (Agencourt AMPure, Beckman Coulter) and cleaned up with 2X SPRI beads. Second PCR with same set of primers was done as follows: 98 °C for 30 s, followed by 6–9 cycles of 98 °C for 10 s, 63 °C for 30 s, 72 °C for 1 min. Each sample was amplified for a total of 11–14 cycles. Libraries were purified with SPRI beads, and concentration was measured by the Qbit dsDNA HS Assay kit (ThermoFischer Scientific) and fragment profiles were analysed with the Bioanalyzer DNA High Sensitivity Kit (Agilent). Libraries were sequenced on 2 × 50 HiSeq 3000 (Illumina) and with an average of 25 million paired-end reads per sample.

**ATAC-seq processing and analysis**. Cutadapt v1.7.1 was used to trim adaptors (http://journal.embnet.org/index.php/embnetjournal/article/view/200). Trimmed paired-end reads were aligned to the mm10 mouse reference genome using Bowtie2 v4.1.2[65] with settings -X 2000 -very-sensitive in paired-end mode. Duplicates were marked with PICARD tools (http://picard.sourceforge.net). Reads were subsequently filtered for alignment quality (>Q30) and were required to be properly paired. Duplicates and reads mapping to the mitochondria, unmapped contigs or chromosomes X and Y were removed. Secondary alignments were not considered. ATAC-seq peaks were called using MACS2 v2.1.1[66] with parameters -nomodel -shift 100 -extsize 200 -keep-dup all -q 0.05. Peaks falling within mouse mm10 ENCODE blacklisted regions (http://mitra.stanford.edu/kundaje/akundaje/release/blacklists/mm10-mouse) were discarded using bedtools v2.24.0 intersect. A consensus peak set of peaks detected in at least two samples was generated using function dba.counts from DiffBind R package v2.6.6 (http://bioconductor.org/packages/release/bioc/htmL/DiffBind.htmL). EdgeR v3.20.9[67] was used to perform a differential accessibility analysis on the set of consensus peaks using DiffBind functions (dba.analyze). *Rxrab*^fl/fl^ LPM-increased chromatin accesibility regions were defined by log2 fold change in read density >1 and

FDR < 0.05. Conversely, *LysM*^Cre+^*Rxrab*^fl/fl^ LPM-increased chromatin accessibility regions were defined by log2 fold change <−1 and FDR < 0.05. Differentially accessible peaks were associated to the gene with the nearest TSS using command annotatePeaks.pl from the HOMER v4.10.3[68] package. UCSC genome browser was used for visualization of ATAC-seq reads in selected genomic regions. Transcription factor motif enrichment analyses for genomic regions showing differential accessibility were performed using HOMER command findMotifsGenome.pl. *LysM*^Cre+^*Rxrab*^fl/fl^ peaks with increased accessibility were compared to *Rxrab*^fl/fl^ LPM-specific peaks and vice versa.

**Ovarian tumour induction and analysis**. Eight to ten-week-old C57BL/6 or *LysM*^Cre+^*Rxrab*^fl/fl^ and *Rxrab*^fl/fl^ female mice, were i.p. injected with 5U of PMSG. Forty-eight hours later, the animals were i.p. injected with 5U of hGC. Thirty-six hours later mice were anesthetized with ketamine/xilacine and 50,000 Upk10 carcinoma cells were injected intrabursally in the left ovary. Upk10 cells were kept in RPMI + 10%FBS + 1% P/S, and injected when reached 70% of confluence. Upk10 cells were derived from primary murine ovarian tumours and kindly donated by Dr Conejo-Garcia (Moffitt Cancer Center, Florida, USA). Upk10 cells were tested mycoplasma negative. C57BL/6 mice were sacrificed 2, 3, 4 or 5 weeks after the Upk10 intrabursal inoculation for tumour growth and leukocyte infiltration monitoring. *LysM*^Cre+^*Rxrab*^fl/fl^ and *Rxrab*^fl/fl^ mice were sacrificed 24 days post-tumoral cell injection. Both ovaries were harvested, and primary ovarian tumours were weighted and measured in three different axis with a caliper for volume calculation using the formula $V = \pi/6 \times L \times W \times H$ (V: volume, L: length, W: width, H: height). For flow cytometry analysis primary ovarian tumour cell suspensions were obtained and stained as described above. For histopathologic analysis primary ovarian tumours were fixed overnight with paraformaldehyde 4%. Tissues were dehydrated in ethanol 70% and included in paraffin for sectioning. Preparations were stained with H&E. Pictures from tissue sections were scanned using NanoZoomer-2.0RS C110730® visualized in NDP.2 viewer.

**Immunofluorescence**. Ovarian tumours were washed with PBS 1X, infused overnight in 5% sucrose, and frozen in OCT. Tumours were maintained at −80 °C and cut by cryostat sectioning into 8-μm slices. The slices were stored at −20 °C, defrosted at RT, and washed three times with PBS 1X. Then, the samples were permeabilized with 0.3% TRITON 100X and 2% bovine serum albumin for 10 min, blocked with 5% goat serum (Jackson ImmunoResearch) for 1 h, labelled with anti-GATA-6 Rabbit mAb (Cell signalling #5851) and anti-F4/80 (CI:A3-1, #ab6640) overnight at 4 °C, labelled with a secondary antibody (goat anti-rabbit AF647) for 1 h at 4 °C, and mounted with Fluromont G mounting media (Southern Biotech). LysoTracker/BODIPY-stained cells were directly imaged using μ-slide eight-well chambers (IBIDI). Images were taken with a Nikon A1R confocal microscope (Nikon) and analysed with Fiji Image J Software.

**Statistical analysis**. All experiments were performed at least twice. Results were statistically analysed with GraphPad Prism 7 using an analysis of variance (ANOVA) test, Mann–Whitney $U$ test, or Student's $t$ test. For all graphical analyses, mean values and S.E.M values were included. For flow cytometry analysis of GATA-6^+^F4/80^+^ macrophages in ovarian tumours we fitted a generalised lineal model with gamma distribution. We used Grubbs' test (GraphPad) to determine significant outliers. A $p$ value of ≤0.05 was considered to be statistically significant.

**Reporting summary**. Further information on research design is available in the Nature Research Reporting Summary linked to this article.

## Data availability
Data supporting the findings of this study are available within the paper and its Supplementary Information files. The source data underlying all figures are provided as a Source Data file, and are further available from the authors upon reasonable request. RNA-Seq and ATAC-Seq data are deposited in GEO; accession number GSE129095 and GSE129414, respectively.

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

## Acknowledgements

We thank the members of the M.M. and M.R. laboratories for extensive discussions and critiques of the paper, and the Immgen Consortium for RNA-sequencing. We thank G. González Pigorini for help with RNA-seq analysis, D. Metzger (Institut de Génétique et de Biologie Moléculaire et Celulaire, Strasbourg, France) for *Rxrb*^fl/fl mice, A. Castrillo (Instituto de Investigaciones Biomédicas "Alberto Sols", Madrid, Spain) for insightful comments, the J.R. Conejo-García laboratory (Moffitt Cancer Center, Florida, USA) for helping with intrabursal injections for the ovarian tumour model, the Genomics Unit at CRG (Barcelona, Spain) for ATAC-seq sequencing, and S. Bartlett (CNIC) for editorial assistance. We also thank the staff at CNIC Microscopy, Cellomics and Animal facilities for technical support. This work was supported by a HFSP fellowship to M.C-A. (LT000110/2015-L/1), grants from the Spanish Ministerio de Ciencia e Innovación (MCI) (SAF2015-64287R, SAF2017-90604-REDT-NurCaMein, RTI2018-095928-B100), La Marató de TV3 Foundation (201605-32) and Comunidad de Madrid (MOIR-B2017/BMD-3684) to M.R, and the Formación de Profesorado Universitario (FPU17/01731) programme (MCI) to J.P. The CNIC is supported by the MCI and the Pro CNIC

Foundation and is a Severo Ochoa Center of Excellence (SEV-2015-0505).

## Author contributions

Conceptualisation: M.C-A., M.P.M-G., M.M. and M.R.; Methodology: M.C-A., M.P.M-G, J.P., D.A-E., Y.L., A.G., S.K., J.L-B., and V.N.; Software: F.W., F. S-C and D.J.C.; Data interpretation and analysis: M.C-A., M.P.M-G., J.P. F.W. and D.J.C.; Writing, reviewing & manuscript editing: M.C-A., M.P.M-G., J.P., M.M., M.R.; Project supervision: M.C-A. M.P.M-G., M.M., M.R.; Funding: M.M. and M.R.

## Competing interests

The authors declare no competing interests.
