## [Peer Review File · Nature Communications]

Reviewers' comments:

Reviewer #1, expert in ovarian cancer (Remarks to the Author):

This paper describes the role of RXRs in the identity of macrophages in serous cavities through regulating chromatin activity and transcriptional regulation in a manner that is partially dependent on retinoic acid signalling. The majority of data described in this paper is of high quality and is a useful addition to the literature and to previous work from these authors.

There are some issues with the ovarian cancer model section. The model is grown in the ovarian bursa and can certainly be considered a model of high-grade serous ovarian cancer in that it has a tp53 mutation which is found in serous cancers, although many such cancers are now thought to arise in the fallopian tube. The information about the model can be found in a reference cited in the methods. The results section should also describe the genetic defects found in these cells. The cells are grown orthotopically but it is difficult to understand how tumor weight could be measured weekly in these orthotopic tumors. This should be described in the methods section. By tumor weight are the authors referring just to the ovarian bursal tumors or any peritoneal metastases as well?

Immune infiltrate numbers in terms of cells/mg tumor should also be given.

Are macrophages with similar phenotypes found in human serous ovarian cancers?

Minor points

The introduction should make it clear that the authors are describing murine macrophage populations. The survival rates of the high-grade serous ovarian cancer type are now improving because of PARP inhibitors (e.g. SOLO 1 trial)

Reviewer #2, expert in tissue macrophages and gene expression (Remarks to the Author):

This is a great paper from the Ricote and Merad groups on the role of RXRa in peritoneal macrophages. The concept that RXRa is not required for the development of the macrophages themselves but is needed for them to be able to deal with lipids is very nice. The effect of the loss of RXRa in macrophages on ovarian cancer development is impressive but some caution would be welcome unless RXRa is not expressed by tumor-associated macrophages. I have two minor comments to be more defensive on the nomenclature and the specific function of LPMs in cancer progression.

Minor comments:

Nomenclature:

This will sound like a lot of semantics but I think it is important not to confuse the readers.

1) In the novel Ms4a3-based monocyte-fate-mapping mice from the Ginhoux lab (Liu et al. *BioRxiv* 2019) Peritoneal macrophages display an increase of Ms4a3-tagging between the 4th week of life and the 12th week of life. I'm not sure this qualifies as "aged" when the authors write that "monocytes contribute to the LPM pool in aged mice". Figure 3 of the Bain et al. *Nature Comm* 2016 paper also shows continuous engraftment between 3 weeks of age and 16 weeks of age. This does not change the fact that the LPM pool is relatively self-maintaining and only has slow and low engraftment of monocytes, but at this moment I think the consensus from these two papers would be that this is slow but continuous and not only in "aged mice".

This also implies that the strict distinction between LPMs as tissue-resident macrophages (I agree) and SPMs as bone-marrow-derived macrophages does not strictly hold up. Part of the LPMs will also be bone-marrow-derived macrophages. Why not stick to LPMs versus SPMs? I then fully agree that all LPMs are tissue-resident macrophages (even if some of them will be coming from monocytes).

2) When the authors knock-down RXRa in the resident LPMs these cells will die and be replaced by bone-marrow derived monocytes (this is supported by the lower percentage of Tim4+ cells, see Bain et al paper *Nature Comm* 2016: macrophages get Tim4 slowly – this is also supported by higher Ccr2

expression in Figure 2A) so that the strict division on TRM versus BDM will not hold up. Even more LPMs will derive from bone-marrow-derived macrophages in the KO animals. Why not keep the LPMs versus SPMs nomenclature as this is now well established in the peritoneal macrophage field?

LyzM-CRE x RXRa-fl/fl:

In these mice all macrophages will be targeted and miss RXRa. It may be that monocyte-derived tumor-associated macrophages also express RXRa. As such Figure 5 could be the effect of loss of most LPMs as proposed by the authors but could also have some effect of the loss RXRa in tumor-associated macrophages. Could the authors be defensive and discuss this in their discussion?

One option would be to sort CD102+ macrophages (infiltrating LPMs) and sort CD102- macrophages (tumor)associated macrophages coming from monocytes that develop into macrophages within the tumor) and compare their expression of RXRa by qPCR. If the tumor-macrophages have low levels of RXRa then we can assume removing RXRa in them will not have too much effect and we can then be more certain that we are mainly looking at the effect on the LPMs. If both macrophages express relatively high levels of RXRa then we need to be a bit more cautious in the discussion of the data.

Please, see below a point-by-point response to referees and editors comments.

Reviewer #1, expert in ovarian cancer (Remarks to the Author):

This paper describes the role of RXRs in the identity of macrophages in serous cavities through regulating chromatin activity and transcriptional regulation in a manner that is partially dependent on retinoic acid signalling. The majority of data described in this paper is of high quality and is a useful addition to the literature and to previous work from these authors.

There are some issues with the ovarian cancer model section. The model is grown in the ovarian bursa and can certainly be considered a model of high-grade serous ovarian cancer in that it has a tp53 mutation which is found in serous cancers, although many such cancers are now thought to arise in the fallopian tube. The information about the model can be found in a reference cited in the methods. The results section should also describe the genetic defects found in these cells.

We thank the reviewer for this comment. The genetic defects of the tumour model has now been described in the Results section, page 10, lines 249-251.

The cells are grown orthotopically but it is difficult to understand how tumor weight could be measured weekly in these orthotopic tumors. This should be described in the methods section.

2, 3, 4 or 5 weeks post Upk10 intrabursal inoculation C57BL/6 mice were sacrificed, and primary ovarian tumour weight and volume were measured with a caliper. We have now clarified this in the Methods section, page 18, lines 507, 511-513, and page 19, line 515. Please see below an image with normal (not injected ovaries) and T (tumours) over time (from 2 to 5 weeks) for kinetics reference (Rebuttal Figure 1).

Rebuttal Figure 1: representative images of orthotopic ovarian tumours (T) and normal ovaries (N) 2, 3 4 and 5 weeks after intrabursal injection of Upk10 cells in C57BL/6 mice.

By tumor weight are the authors referring just to the ovarian bursal tumors or any peritoneal metastases as well?

Tumour weight/volume measurements and flow cytometry analysis were performed in primary ovarian bursal tumours. We have clarified it in the Results (page 10, lines 251-253, and 261-262) and Methods section (page 18, line 514, and page 19, lines 516 and 518), as well as in Figure 5 (page 29, lines 675, and 680), Supplementary Figure 5 (page 10 of the Supplementary Information file, lines 92, and 94) and Supplementary Figure 6 legends (page 11 of the Supplementary Information file, lines 104, and 109). We detected metastases at multiple peritoneal locations in a proportion of C57BL/6 animals (around 25-30%) from 4 weeks after the Upk10 injection on. Of note, tumour analysis in *LysM^{Cre+}Rxrab^{fl/fl}* and *Rxrab^{fl/fl}* mice were performed 24 days after tumor inoculation, when peritoneal metastases were barely detected.

Immune infiltrate numbers in terms of cells/mg tumor should also be given.

Immune infiltrate numbers in terms of cells/mg tumour, and frequency of cells among CD45+ leukocytes have been now included in Supplementary Figure 5.

Are macrophages with similar phenotypes found in human serous ovarian cancers?

We thank the reviewer for this interesting question. There is a study comparing the transcriptomes of TAMs from human ovarian carcinoma ascites with peritoneal macrophages from patients undergoing hysterectomy for non-malignant diseases (pMPH), and monocyte-derived macrophages (MDM) (1). The authors of this study found high similarity between TAMs and resident peritoneal macrophages, considerably greater than the resemblance of TAMs and monocyte-derived macrophages. They found that human ovarian carcinoma ascites expressed markers commonly expressed in mouse LPMs, including GATA-6, ADGRE1 (F4/80) and TIMD4 (1). Interestingly, we have analysed their RNAseq data and found that RXR α is expressed in TAM and pMPH (Rebuttal Figure 2). We have now expanded on this finding in the Discussion section, page 12, lines 319-322.

Rebuttal Figure 2: Expression of genes coding for LPM-specific markers and RXR α in TAMs from human ovarian carcinoma ascites, human peritoneal-resident macrophages (pMPH), and human monocyte-derived macrophages (MDM), from published RNA-sequencing data (1).

1. Finkernagel F, *et al.* The transcriptional signature of human ovarian carcinoma macrophages is associated with extracellular matrix reorganization. *Oncotarget* 7, 75339-75352 (2016).

Minor points

The introduction should make it clear that the authors are describing murine macrophage populations

This is now stated in the Abstract (page 2, lines 36, and 41), the Introduction section (page 3, line 59, and page 4, lines 85, and 89), and the Discussion section (page 12, line 320).

The survival rates of the high-grade serous ovarian cancer type are now improving because of PARP inhibitors (e.g. SOLO 1 trial)

We thank the reviewer for his/her comment. We have now included this piece of information in the Discussion section, page 12, lines 315-316.

Reviewer #2, expert in tissue macrophages and gene expression (Remarks to the Author):

This is a great paper from the Ricote and Merad groups on the role of RXRa in peritoneal macrophages. The concept that RXRa is not required for the development of the macrophages themselves but is needed for them to be able to deal with lipids it very nice. The effect of the loss of RXRa in macrophages on ovarian cancer development is impressive but some caution would be welcome unless RXRa is not expressed by tumor-associated macrophages. I have two minor comments to be more defensive on the nomenclature and the specific function of LPMs in cancer progression.

Minor comments:

Nomenclature:

This will sound like a lot of semantics but I think it is important not to confuse the readers.

1) in the novel Ms4a3-based monocyte-fate-mapping mice from the Ginhoux lab (Liu et al. BioRxiv 2019) Peritoneal macrophages display an increase of Ms4a3-tagging between the 4th week of life and the 12th week of life. I'm not sure this qualifies as "aged" when the authors write that "monocytes contribute to the LPM pool in aged mice". Figure 3 of the Bain et al. Nature Comm 2016 paper also shows continuous engraftment between 3 weeks of age and 16 weeks of age. This does not change the fact that the LPM pool is relatively self-maintaining and only has slow and low engraftment of monocytes, but at this moment I think the consensus from these two papers would be that this is slow but continuous and not only in "aged mice".

This also implies that the strict distinction between LPMs as tissue-resident macrophages (I agree) and SPMs as bone-marrow-derived macrophages does not strictly hold up. Part of the LPMs will also be bone-marrow-derived macrophages. Why not stick to LPMs versus SPMs? I then fully agree that all LPMs are tissue-resident macrophages (even if some of them will be coming from monocytes).

2) When the authors knock-down RXRa in the resident LPMs these cell will die and be replaced by bone-marrow derived monocytes (this is supported by the lower percentage of Tim4+ cells, see Bain et al paper Nature Comm 2016: macrophages get Tim4 slowly – this is also supported by higher Ccr2 expression in Figure 2A) so that the strict division on TRM

versus BDM will not hold up. Even more LPMs will derive from bone-marrow-derived macrophages in the KO animals. Why not keep the LPMs versus SPMs nomenclature as this is now well established in the peritoneal macrophage field?

We thank the reviewer's comment. We have now included Liu et al.,2019 in our bibliography and changed the Introduction section to state that "monocytes slowly and continuously contribute to the LPM pool", and not only in aged mice as previously stated (page 3, lines 68-70). In order to avoid any confusion related to cell ontogeny we have now used the LPM/SPM nomenclature in the revised version of the manuscript.

LyzM-CRE x RXRa-fl/fl:

In these mice all macrophages will be targeted and miss RXRa. It may be that monocyte-derived tumor-associated macrophages also express RXRa. As such Figure 5 could be the effect of loss of most LPMs as proposed by the authors but could also have some effect of the loss RXRa in tumor-associated macrophages. Could the authors be defensive and discuss this in their discussion? One option would be to sort CD102+ macrophages (infiltrating LPMs) and sort CD102- macrophages (tumor)associated macrophages coming from monocytes that develop into macrophages within the tumor) and compare their expression of RXRa by qPCR. If the tumor-macrophages have low levels of RXRa then we can assume removing RXRa in them will not have too much effect and we can then be more certain that we are mainly looking at the effect on the LPMs. If both macrophages express relatively high levels of RXRa then we need to be a bit more cautious in the discussion of the data.

We agree with the reviewer that RXR-expressing monocyte-derived macrophages might contribute to the phenotype observed in tumour growth when using *LysM^{Cre+}Rxrab^{fl/fl}* mice due to LysMCre expression on monocytes. This is now stated in the Results section, page 10, lines 271-272. As suggested, we have now isolated TAMs (F4/80+CD11b+CD102^{NEG}) and LPMs (F4/80+CD11b+CD102^{POS}) from early ovarian tumours (day 24 post-tumor implantation) and quantify *Rxra* mRNA expression on these cells (Rebuttal Figure 3). We found that both TAMs arising from monocyte precursors and LPMs infiltrating the ovarian tumours express *Rxra* transcripts, the latter displaying higher levels (Rebuttal Figure 3). Nevertheless, our data showed that monocytes (Figure S1d) and monocyte-derived cells within ovarian tumors (Figure S6e) are not affected in RXR-deficient mice, supporting that RXR deletion in LPMs is responsible for the effect observed on tumour growth.

Rebuttal Figure 3. A) Flow cytometry plots from naïve and Upk10 ovarian tumours analyzed at day 24 post-tumour injection. Cells were gated as Singlets DAPI^{NEG}B220^{NEG}CD3^{NEG}Ly6G^{NEG}F4/80⁺ and sorted based on CD102 expression. **B)** Quantitative PCR levels (represented as Delta Ct for absolute quantification) of *Rxra* transcripts for F4/80⁺CD102^{NEG} TAMs and F4/80⁺CD102⁺ LPMs.

REVIEWERS' COMMENTS:

Reviewer #1 (Remarks to the Author):

The authors have answered my questions and this has improved and clarified some aspects of the paper.

Reviewer #2 (Remarks to the Author):

I thank the authors for addressing my comments and congratulate the Ricote team and the Merad team for beautiful work.